# What Is the Relationship between Trunk Control Function and Arm Coordination in Adults with Severe-to-Moderate Quadriplegic Cerebral Palsy?

**DOI:** 10.3390/ijerph20010141

**Published:** 2022-12-22

**Authors:** María Isabel Cornejo, Alba Roldan, Raul Reina

**Affiliations:** 1Sports Research Centre, Department of Sport Sciences, Miguel Hernández University of Elche, 03202 Elche, Spain; 2Exercise and Rehabilitation Sciences Institute, School of Physical Therapy, Faculty of Rehabilitation Sciences, Universidad Andres Bello, Santiago 7591538, Chile

**Keywords:** manual dexterity, intra-limb coordination, brain impairment

## Abstract

Adults with tetraparesis cerebral palsy (i.e., wheelchair users) tend to experience more accelerated ageing, resulting in physical deterioration that increases the impact of the disability, leading to a loss of mobility that interferes with people’s daily activities and participation in the community. The aim of this work is to study the relationship between trunk control and the function of the less-affected arm in this population. For this purpose, 41 para-athletes were invited to participate in this study, performing five tests to assess upper limb coordination, two tests to assess manual dexterity [i.e., Box and Block Test (BBT) and Box and Ball Test (BBLT)] and three tests to assess intra-limb coordination in different planes. Trunk control was assessed in both static and dynamic sitting conditions. The results show moderate correlations between static postural control and manual dexterity tests in the BBT (*r* = −0.553; *p* = 0.002) and BBLT (*r* = −0.537; *p* = 0.004). Large correlations were also found between static postural control and intra-limb tasks in horizontal (*r* = 0.769; *p* = 0.001) and vertical movements (*r* = 0.739; *p* = 0.009). Better static trunk control is related to a better upper limb function in the sagittal plane. Considerations and implications are explained in the manuscript.

## 1. Introduction

Quadriplegic cerebral palsy (CP) is the most severe form of this health condition in which people with such involvement are usually classified as Level III or IV on the Gross Motor Function Classification Scale (GMFCS), presenting a severe involvement of motor function that generally necessitates the use of power wheelchairs for daily activities [1]. Motor issues in CP tend to affect trunk control and upper limb function, but their relationship has been studied principally in children and young people with CP [2,3]. Kusumoto et al. [4] reported that better trunk control and, therefore, greater proximal stabilisation could improve upper limb function, suggesting a positive impact on self-care tasks in children and adolescents with spastic CP. However, little is known about this relationship in adulthood. The literature reported that the higher the GMFCS level, the more sedentary the lifestyle of the individual [5], leading to a more progressive reduction of motor function [6]. Unfortunately, being active with quadriplegic CP is not an easy task because there are limited activities available [7].

Boccia is a strategy and precision para-sport, where several balls are thrown with the aim of getting closest to a target (i.e., the jack). This physical activity promotes practical opportunities for people with severe physical impairments (i.e., high support needs) and was initially designed for people with quadriplegic CP, whose four limbs and control trunk are affected [8]. Playing boccia requires high concentration and tactical skills, as well as physical and manipulative skills, such as gripping and releasing movements [9]. Sensory information processing, movement planning and motor control are relevant factors for proper upper limb throwing, manual dexterity (MD) and coordination skills, which are components that are often affected in people with neurological impairments [10].

In boccia, motor coordination is necessary for performance, which can be defined as the ability to execute fast and skilled movements with fluency, speed, and accuracy [11]. In this para-sport, MD and intra-limb coordination (ILC) are two of the primary skills assessed during the classification process, owing to the relevant impact on sport-specific activity limitations [8,12]. MD requires neuromuscular and neurological aspects to produce precise movements, achieving the ability to grasp and manipulate objects [13]. People with CP tend to present an altered function of the hand that impacts fingers and wrist movements, limiting the execution of precise activities in gripping and manipulation tasks [14], which are also necessary for many activities of daily life [15,16].

One of the most used tools in the clinical setting for assessing MD is the Box and Block (BB) test [17]. This test evaluates the ability to grasp and release objects, an appropriate test in the assessment of adults with neurological conditions [18], such as a prior stroke [19,20] and CP [14,21]. On the other hand, to assess specific throwing coordination, Connick et al. [11] proposed using different tapping tests to identify the impact of the coordination impairment on the activity limitation. Tapping tests are capable of assessing accurate temporal movements and ILC between the shoulder, elbow, forearm, wrist and hand movements in the same limb [10]. Roldan et al. [12] proposed using a battery of tapping tests with discrete and continuous tasks that allow the evaluation of impaired coordination in individuals with severe-to-moderate impairments, considering the movements of the upper extremities and the characteristics of the objects manipulated (e.g., boccia balls).

Trunk control is also an essential component in adults with CP with severe-to-moderate impairments. A recent study by Barbado et al. [22] demonstrated that people with CP present greater problems in trunk control than those without a disability. It has also been shown that people with CP have limited ability to maintain appropriate postural control during the practice of this para-sport [9]. Postural control is considered a sensorimotor function that requires maintaining body position to promote orientation and stability during motor tasks [23], whereas trunk control requires stabilisation and the performance of selective movements of the trunk [24]. For trunk control assessments, a reliable battery of posturographic tests in static and dynamic conditions was proposed [22], demonstrating that dynamic tests on a stable surface are feasible and adequate for identifying different degrees of severity in people with CP and quadriplegic involvement.

According to Cankurtaran et al. [25], the principal factors that predict the sitting function in non-ambulatory children with CP are trunk control and upper extremity function. A relationship between trunk stability and reaching activities was reported, where participants with less postural control showed a body trajectory (i.e., the centre of pressure (CoP)) with multiple disturbances when requested to perform an aiming task [26]. On the other hand, those with better postural stability showed better performance in coordinated reaching tasks [27]. Postural dysfunctions can also be related to the degree of disability, the ability to perform daily living activities [28] and self-care function in people with CP [29]. From the physical-activity practice perspective, the function of the trunk can be relevant because individuals who could reach long distances in boccia (9–10 m) had better performance [30].

A better understanding of trunk function in people with CP can influence the ability to perform upper limb activities. To the best of the authors’ knowledge, no study has explored the relationship between impaired upper limb coordination and postural control, so this study aims to explore this relationship in a sample of boccia para-athletes.

## 2. Materials and Methods

### 2.1. Participants

A cross-sectional study was conducted with a group of 41 boccia players with CP (35.2 ± 15.1 years) presenting severe-to-moderate physical impairments. All of them were classified as Level III or IV on the GMFCS and in sports classes BC1 (*n* = 16) and BC2 (*n* = 25) for boccia para-sport. Participants were recruited from rehabilitation centres and sports clubs where people with neurological impairments train and play boccia. Participants were informed that participation was voluntary and that they had the right to withdraw at any time if they were not comfortable. The inclusion criteria were (i) having a brain impairment from CP or a similar, eligible neurological condition; (ii) classified as BC1 or BC2; (iii) having an active boccia licence; (iv) having no surgeries or botulinum toxin A injections in the six months before testing; and (v) able to properly follow the test instructions. The exclusion criterion was having any comorbidity of intellectual impairment that would impede the ability to follow the testing instructions. Ethical approval was obtained through the local university ethics committee (Reference no. DPS-RVV-001-10).

### 2.2. Procedures

All the participants performed tests to assess MD, ILC and trunk control in a single session, performing coordination tests afterwards to assess trunk control.

#### 2.2.1. Manual Dexterity Tests

BB test and Box and Ball (BBL) tests were used to assess impaired MD. The materials used for the BB test evaluation were a wooden box (53.7 × 25.4 cm) divided into two compartments by a partition 15.2 cm in height and wooden blocks measuring 2.5 cm per side. For the test, the application used the procedure described originally by Mathiowetz et al. [31]. The box was placed on a height-adjustable table, and participants were seated in their wheelchairs in front of it. In this position, the participants had to transport the blocks from one compartment of the box to the opposite side as quickly as possible, with the aim of transporting as many blocks as possible. The test was performed with the throwing hand for para-sport and the preferred hand for daily living activities. Participants were allowed 10 s to obtain a familiarisation with the test and then completed two attempts for 1 min each, with 1 min rest between trials.

The BBL test procedure was similar, but six boccia balls replaced the blocks. This procedure was described previously by Roldan et al. [12] as an alternative way to measure MD using sport-specific materials in boccia para-athletes with CP.

Test outcomes were recorded with a video camera (Sony HDR-PJ410B, Tokyo, Japan) on a tripod (Hama Star 63, Monheim, Germany) located 2 m in front of the participants to subsequently identify the performance of the task by counting the number of blocks and balls passed. A stopwatch (Casio HS-30W-1V, Tokyo, Japan) was used to measure the time lapsed for completing both tests.

#### 2.2.2. Intra-Limb Coordination Test

The ILC was assessed with three tapping tests previously described by Connick et al. [11] and Deuble et al. [32], showing good reliability and validity in ambulant para-athletes with brain/coordination impairments (i.e., ICC > 0.80). The tests used were the Discrete Horizontal Finger Tapping (DHFT), Discrete Vertical Tapping with Ball (DVTB) and Continuous Vertical Tapping with Ball (CVTB) tests. The average movement time of the arm (in seconds) was the measurement used for the discrete tests, whereas the continuous test evaluated the number of taps that each participant was able to perform between the plates for the duration of the test.

Two metal plates (i.e., A and B) placed side by side were required for the DHFT. Each plate was 30 cm long by 20 cm wide. The target area, located in the centre of each plate, was 18 cm long by 5 cm wide. To perform the test, participants had to complete 10 tapping cycles. A cycle was completed when the participant moved their finger from Plate A to Plate B (finish position) as fast as possible. The participant had to receive a “go” signal from the evaluator to start each cycle. The tapping was performed with the participant sitting in their wheelchair with their throwing hand closed and index finger extended, while holding the non-throwing arm across the chest.

The platforms were placed on a height-adjustable table, and the participant was positioned parallel to it. The plate table was adjusted so that the bottom of the table aligned with the individual’s hips (greater trochanter) and the shoulder of the player’s throwing arm was aligned with Plate A (start position). For the correct performance of the test, all the participants were positioned in the same way. This test was designed to assess how fast an individual can move their finger from one plate to the other, and performance was measured by the average time of the 10 tapping cycles (in seconds). The interval between tapping trials was set for a period of at least 3 s.

The two tapping tests that assessed movement in a vertical orientation required an L-shaped (90°) platform, where the contact plate was kept horizontal, but Plate B was placed on the vertical edge. The distance between the plates’ centres was 30 cm. In the DVTB test, the participant was positioned on one side of the platform such that the shoulder was aligned with the plates, allowing shoulder flexion–extension movements. The objective of the test was to measure how fast the para-athlete could move their finger from one platform to the other, recording the mean time taken to perform the 10 cycles requested.

The other vertical tapping test was the CVTB test, where participants sat in front of the testing table with their less-affected shoulder (or the one used for throwing/aiming actions) located in the centre of Plate A, 30 cm from the edge of Plate B. Contact could be made at any point on the plates (14 × 17 cm). Participants used a boccia ball to make contact between the plates and perform continuous movements, consisting of grasping the ball and tapping Plates A and B alternately as fast as possible in 1 min. The total number of contacts made was recorded. Two attempts were made, with a rest period of at least 60 s between trials. The evaluator gave the go and stop signals to start and end the test, respectively.

In all the tapping tests, an electrical impulse (i.e., contact on the plates) was recorded with an A/D converter (USB-6001; National Instruments, Austin, TX, USA). The data from the A/D converter were recorded with a program developed within LabVIEW^®^ 2009 software (version 2.04; National Instruments, Austin, TX, USA).

#### 2.2.3. Trunk Control Test

To assess trunk control, participants performed different tests using a protocol designed by Barbado et al. [33]. This protocol considers performing trunk tasks on stable and unstable sitting surfaces; however, in this study, the trunk control tasks (static and dynamic) were performed on a stable seat (see Figure 1 for reference).

An iron structure for measuring trunk control was built with a stable and flat wood surface at a height of 0.9 m to place the force plate (Kistler 9286AA, Winterthur, Switzerland). The force plate was sampled at 1000 Hz. All participants were positioned in the same way and placed on the seat with knees at 90° and legs tied to the seat in order to prevent lower limb movements.

A screen (106 × 138 cm) was placed 3.5 m in front of the participant, where real-time visual biofeedback of the CoP displacement (Hitachi CP-X300, Ibaraki, Japan) was projected. A yellow dot (radius of 60 mm) was shown on the screen and represented the displacement of the CoP achieved during the test by the participant. Additionally, a red dot (i.e., a target point) was presented to participants in several tasks to assess their ability to adjust their CoP position to the target location.

Five posturographic tests were applied: two static and three dynamic tasks. The first static condition was without feedback, and participants were asked to maintain their upright sitting posture as still as possible while the target point was not presented on the screen. The second static condition was similar but provided visual feedback on the screen. During the dynamic tasks (with visual feedback), participants were instructed to follow the target in red, which moved in anterior–posterior, medial–lateral and circular trajectories. In the dynamic tasks, the target point took 20 s to complete a cycle, moving at a rate of 0.005 Hz [22,33].

The five tests were performed in order of ascending difficulty. Each posturographic task was performed twice, with a testing duration of 70 s and a rest time of 60 s. To consider a trial successful and proceed to the next task, the participant had to complete the test with two or fewer instances of external assistance or <15 s of the 70 s total testing time. The test would stop when the participant was unable to complete the second trial of the task or when the second trial finished correctly, completing the two trials of the five tasks [34].

### 2.3. Posturography Data Reduction

To filter the CoP signal, a low pass filter was used (4th-order, zero phase lag, Butterworth, 5 Hz cut-off frequency), while the CoP time series were subsampled at 20 Hz, and the first 10 s of each trial were not considered for data analyses to avoid non-stationarity information related to the beginning of the trial [22]. To quantify trunk control during the trials, the mean radial error (MRE) was calculated as the average vector distance (mm) of the CoP from the target point (i.e., the participant’s mean CoP position) [35], where a larger MRE means more trunk sway, that is, worse performance or trunk function. The best trial performed for each condition (lowest MRE) was used for the statistical analyses. The static (task with visual feedback + task without visual feedback) and dynamic (task with anterior–posterior + medial–lateral + circular trajectories) composite scores were also reported as the mean ± standard deviation and used for the correlation analysis.

### 2.4. Statistical Analysis

Descriptive statistics were generated for all the participants and presented as means and standard deviations. Pearson’s product-moment correlation coefficient (*r*) was used to analyse the strength of association between the results of the ILC and MD tests with the posturographic battery. Threshold values were used to interpret the correlation coefficients: <0.1, trivial; 0.1–0.3, small; 0.3–0.5, moderate; 0.5–0.7, large; 0.7–0.9, very large; and >0.9, nearly perfect [36]. Data analysis was performed using the Statistical Package for Social Sciences (SPSS Inc., version 23.0 for Windows, Chicago, IL, USA), with GraphPad version 5 (San Diego, CA, USA) to produce the graphs.

## 3. Results

Table 1 shows the mean scores obtained in each task of the MD, ILC and trunk control tests, evidencing that more balls than blocks were transferred from one box to the other after the MD testing period; on average, more time was required to perform vertical tapping contacts than horizontal; and a larger MRE was exhibited in dynamic versus static trunk control tests.

Table 2 shows the relationship between the MD and trunk control tests, showing a moderate negative significant correlation between the BBL (*r* = −0.553; *p* = 0.002) and the BB (*r* = −0.537; *p* = 0.004) tests. In the dynamic conditions, the relationships were small and negative for the BB (*r* = −0.405; *p* = 0.032) and the BBL (*r* = −0.290; *p* = 0.142) tests. The associations between the trunk control and ILC tests can also be observed in Table 2. Large positive correlations were found between the static control test and the DHFT (*r* = 0.769; *p* = 0.001) and the DVTB (*r* = 0.739; *p* = 0.009) tests, while the association with the CVTB test was moderate and negative (*r* = −0.616; *p* = 0.044). Regarding the dynamic trunk control tasks, only the DHFT test had a positive moderate association (*r* = 0.677; *p* = 0.006). For the other two ILC tasks, the DVTB test had a moderate but non-significant correlation (*r* = 0.529; *p* = 0.094), and the CVTB test had a small negative and non-significant correlation (*r* = −0.120; *p* = 0.726). The plots of the associations found are included in the Appendix A of this manuscript.

## 4. Discussions

This study aimed to evaluate the relationship between impaired upper limb coordination and postural control in individuals with high support needs. Our results show that the trunk function and upper limbs are related in adults with severe-to-moderate CP. This study approached upper limb proficiency from two perspectives: MD and ILC. From the first perspective, our results show that individuals who performed better in MD tasks presented more stable trunk control during activities that required fine hand and finger movements. A recent study showed similar results in children with less impairment (i.e., hemiplegia), suggesting that hand function might be improved by training in core stability exercises [3].

Concerning the ILC tasks, our results show that the dynamic trunk control test had moderate correlations with the DHFT test. It is plausible to think that those who have better trunk control in dynamic conditions have more skills and proficiency in performing movements in the sagittal plane (i.e., flexion and extension movement of the trunk). Previous studies that analysed trunk control during the performance of dynamic tasks in different planes of movement showed that people with CP had less difficulty reaching or moving in the sagittal plane, as opposed to the coronal and/or transverse plane [37]. The sagittal plane is indeed important for performing anterior reaching tasks in this population, so motor interventions must be designed to take this fact into account. Additionally, non-ambulant individuals with CP tended to perform worse in the transversal plane because the wheelchair configuration and lateral armrests limited movements on both sides of the body, regardless of trunk function. This situation can cause the trunk musculature to become weakened through disuse so that it cannot be used properly outside the body. When examining ILC, the DHFT and DVTB tasks showed that those participants who spent more time performing the requested coordination tasks had a lower performance in the static trunk tests. In contrast, in the CVTB task, the relationship was negative, considering that those participants who made more contacts had lower scores in the trunk tests, that is, better trunk control performance. Therefore, to perform proficiency activities that required coordination with the upper limb, it could be more relevant to have had a stable trunk function. This is similar to the results described by Roldan et al. [34], which showed that static trunk control could be more relevant than dynamic trunk control. Boccia requires that individuals throw balls accurately, so they may not need broad trunk movements but compensatory strategies to stabilise themselves and achieve the most optimal throwing position [38]. In this respect, the hip-, pelvis- and trunk-support devices used by people with severe CP, along with the impairment itself, can reduce the number of degrees of freedom in joint movements, which must be controlled by reducing the demand of the postural control task [39] or making it possible for movements to occur with less variability or error. Therefore, it is interesting to consider the strategies that people with severe-to-moderate CP use to maintain a stable trunk, thus allowing the practice of activities that require DM and ILC. These are called compensatory strategies and, together with the wheelchair configuration, can influence the individual’s performance of the tasks.

To reinforce the positive influence of external support in trunk stabilisation, Santamaria et al. [40] reported that adequate external trunk support facilitates improved motor performance and thus facilitates daily living tasks such as manipulative activities or self-care in participants with severe CP [41], similar to the participants of this study. As a practical or technical contribution of this study, it can be highlighted that to optimise fine motor function and upper limb intra-limb coordination in adults with severe-to-moderate CP, it is necessary to consider static trunk control when performing aiming tasks. Dynamic trunk control can also be relevant, mainly in reaching tasks involving movements in the sagittal plane.

Some limitations should be considered in this research. The number of participants was limited, mainly owing to the transportation difficulties and daily living requirements of people with high support needs. As a consequence, the severity of the disabilities of boccia athletes also limited the completion of the trunk control test battery, with some of the athletes only able to complete the static tasks of the protocol.

## 5. Conclusions

This study highlights that, in people with severe-to-moderate CP (i.e., boccia para-athletes with high support needs), static trunk control is relevant during the performance of tasks requiring coordination and fine motor skills with the upper limb. This is in contrast to dynamic trunk control, which seems to be less relevant to the performance of this type of task. Therefore, the relationship of MD and ILC with trunk control can be an interesting aspect to consider during participation in boccia para-sport. In addition, MD and ILC can be relevant components during the activities of daily living in these groups. Further research can explore the optimal position in the wheelchair and the most used compensatory strategy that contributes to a stable posture and, thus, an efficient manipulative function in daily life and sports activities.

Future research can address this topic in adults and/or older people with severe-to-moderate CP, comparing subgroups according to their sports participation (e.g., boccia players or para-athletes with high support needs vs. non-athletes or sedentary individuals), to determine if boccia para-sport can have a positive implication on upper limb and/or trunk functions.

## Figures and Tables

**Figure 1 ijerph-20-00141-f001:**
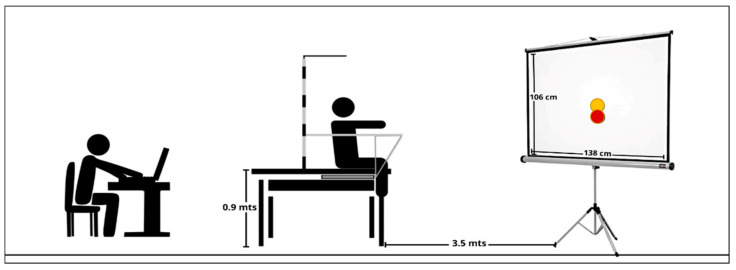
Settings for trunk control test throughout posturography on stable surfaces (red point = target point; yellow dot = CoP displacement).

**Table 1 ijerph-20-00141-t001:** Means and standard deviations of the manual dexterity (MD), intra-limb coordination and trunk control tests.

Task	Mean		SD
MD tests			
BB (N blocks)	22.78	±	8.52
BBL (N balls)	30.89	±	12.35
ILC tests			
DHFT (s)	896.07	±	544.40
DVTB (s)	969.82	±	607.96
CVTB (N contacts)	39.47	±	16.99
Trunk control tests			
Static (mm)	4.67	±	3.66
Dynamic (mm)	9.59	±	4.52

SD = standard deviation; MD = manual dexterity; BB = Box and Block test; BBL = Box and Ball test; ILC = intra-limb coordination; DHFT = Discrete Horizontal Finger Tapping test; DVTB = Discrete Vertical Tapping with Ball test; CVTB = Continuous Vertical Tapping with Ball test.

**Table 2 ijerph-20-00141-t002:** Pearson product correlation between intra-limb coordination and static and dynamic trunk control tests.

Task	Static Control Test		Dynamic Control Test	
	*r*	*p*	*r*	*p*
MD tests				
BB (N blocks)	−0.553	0.002 **	−0.405	0.032 *
BBL (N balls)	−0.537	0.004 **	−0.290	0.142
ILC tests				
DHFT (s)	0.769	0.001 **	0.677	0.006 *
DVTB (s)	0.739	0.009 *	0.529	0.094
CVTB (N contacts)	−0.616	0.044 *	−0.120	0.726

BB = Box and Block test; BBL = Box and Ball test; ILC = intra-limb coordination; DHFT = Discrete Horizontal Finger Tapping test; DVTB = Discrete Vertical Tapping with Ball test; CVTB = Continuous Vertical Tapping with Ball test. * *p* < 0.05; ** *p* < 0.01.

## Data Availability

All relevant data are within the paper.

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
