# Peer review of "What Is the Relationship between Trunk Control Function and Arm Coordination in Adults with Severe-to-Moderate Quadriplegic Cerebral Palsy?"

_ijerph, 2022, doi:10.3390/ijerph20010141_

Round 1

Reviewer 1 Report

- Topic is crisp but he paper needs to explicitly state or clarify the following questions: (a) what are the underlying challenges? (b) what are the technical contributions of this paper?
It is not at all clear from the paper the 'novelty' or the 'contribution' of this research.
- High plagiarism detected 42 % by Turnitin (maybe it can vary but I am attaching the report for you)
- Huge background study should be needed
- Can be accepted with above revision as the content is interesting 

Reviewer 2 Report

The article is interesting, understandable and suitable for the special issue "The influence of physical activity, sedentary behavior and fitness on cognitive functions and well-being".

The abstract gives a good insight into the study.

The purpose of the article is clearly stated. In the introductory part, the framework and problems faced by quadriplegics with cerebral palsy are adequately explained, as well as the importance of physical activity for this extremely vulnerable group of people with limited opportunities for sports. A detailed description of the methods is a particularly strong part of the study. Very well presented. The results are clearly displayed. The discussion focused on the topic without over-expanding ideas. The conclusion is logically derived from the obtained results with a clearly highlighted potential contribution of this research to improving the quality of life of people with severe disabilities.

I suggest to the authors that, in addition to the graphical presentation, they should also present the results in a table.

Compliments to the authors.

Round 2

Reviewer 1 Report

- Everything looks good and revised well as instructed in the previous review.